# Antecedents of Creating Business Models in the Field of Renewable Energy Based on the Concept of the New Age of Innovation

**Jan Brzóska [1], Lilla Knop [1], Monika Odlanicka-Poczobutt [1] and Dagmara K. Zuzek [2],***

[1] Department of Organization and Management, Silesian University of Technology, 44-100 Gliwice, Poland; jan.brzoska@polsl.pl (J.B.); lilla.knop@polsl.pl (L.K.); monika.odlanicka-poczobutt@polsl.pl (M.O.-P.)

[2] Department of Economics and Food Economy, Agricultural University in Krakow, 31-120 Kraków, Poland

* Correspondence: d.zuzek@urk.edu.pl; Tel.: +48-695-550-850

**Abstract:** The purpose of this paper is to identify the antecedence of business models for the renewable energy sector, characterise their concepts and structure, and assess the importance of innovation in the creation of value for the customer and for the business in the examined business models. According to the concept of K. Prahalad and M. S. Krishnan, an operational business model consists of three basic components. The first two are social architecture and technical architecture that represent specific resources. The third component comprises business processes. The operating business model according to the principles of the New Era of Innovation was used. The business models were presented as case studies of the following: a photovoltaic farm, a passive building and a local (communal) biogas plant. A desk research method was employed as well as triangulation of the research methods of non-participant observation, unstructured interview and business documentation review, in order to develop the case studies. The identification of antecedents for business models in renewable energy generation allows the argument that the business model can be treated as a system for the configuration of resources and interdependent activities, emphasising the role of the configuration of tangible and intangible elements. As the presented business models have been active for a relatively limited time, changes in competence and human capital attitudes, as well as social acceptance for those models, are not examined here. The absence of the upper and lower performance limit of the optimisation algorithm, or system variables, may be an interesting area for further research.

**Keywords:** renewable energy; business models; The New Age of Innovation; sustainability development

## 1. Introduction

The energy sector is a major infrastructural sector that enables socio-economic and civilisational development of countries and regions. The requirements of sustainable development cause the energy sector, which exerts a major impact on the natural environment and fuels climate change, to undergo a major technology and organisational transition, resulting from the implementation of various innovative solutions. They promote changes in business models employed by energy undertakings and generate new, innovation-dominated business models.

Business models of energy undertakings are the object of many discussions. Currently, on the back of the growing importance of RES (renewable energy sources), business models of undertakings that apply those solutions deserve special attention. More importantly, as reported by the Ember and Agora Energiewende think-tanks, in 2020, *renewable energy sources* (RES), for the first time ever, were the main source of electricity in the European Union. Disappointingly, Poland (17%) is at the bottom of the list of countries using RES for electricity production, with only Czechia (12%) and Hungary (15%) behind. To note, the consumption of fossil fuels in electricity production in Poland is as much as 83%, the

highest in the EU [1]. The above notwithstanding, the awareness of using RES is ever rising, producing case studies of undertakings operating in the Polish market.

The purpose of this paper is to identify the antecedents of business models for the renewable energy sector, characterise their concepts and structure, and assess the importance of innovation in the creation of value for the customer and for the business in the examined business models. The research topic was the development and application of a methodology to examine the structure of innovative energy sector business models, to allow the evaluation of their capacity to create multi-dimensional value, and the comparative analysis of the examined business models in terms of their applicability in the selected economic sectors and public domain. The following questions were formulated for the research:

Q1: What are the most important antecedents for the development of innovative business models in the RES-based energy sector?

Q2:What particular elements and properties of business models can be identified in the examined cases?

Q3: How can the methodology be developed to examine business models in terms of value creation?

Antecedents for the creation of new business models in the energy sector are rooted in climate policy, in particular energy policy [2], laws and regulations [3], the limited nature of fossil energy carriers [4], robust growth of innovation [5], environment-friendly social activities [6] and digital economy growth, in particular the smart power grid, digital energy security facilities, development of innovative technologies and equipment for the renewable energy sector [5]. Environmental discontinuity and turbulence, crises and the imperative of sustainable development all result in the application of new management systems in business operations. Businesses are looking for effective methods to compete and improve their efficiency, whilst respecting the principles of sustainable development. This results in the development and implementation of new business strategies and models, in which innovation and collaboration are used to an ever greater extent. In the case of energy undertakings, a major factor in their strategic reorientation, or in the establishment of new organisations, is the so-called Green Deal Policy, notably the part that deals with energy policy and innovation development strategy as implemented by the state. In this context, there is a growing interest on the part of management theorists and practitioners in business models of energy undertakings. The major factors underlying such a significant interest in this domain are:

- application of the business model as a transparent vehicle for the creation of value, for the customer, prosumer and business owner alike [7];
- the business model as a creator and carrier of various types of innovation. Process innovation (new energy generation technologies) and marketing innovation (relations with customers and prosumers) are of particular importance in the energy sector at present [8–11];
- the search for instruments and methods of obtaining competitive advantages. In the case of the energy sector, the cost advantage is particularly important, although quality aspects (e.g., reliability of energy supply, environment-friendly energy generation) [12] are gaining in importance;
- treatment of the business model as the architecture for business activities, capable of generating value that is the basis for income generation and contribution to energy security [13];
- business models of the innovative energy sector address the requirements of the EU energy and climate policy [2,3];
- use of the business model as a vision of the business idea representing a proposition for prospective investors and lenders [14].

Business models applied by three innovative economy undertakings that use renewable energy sources, i.e., PV (*photovoltaics*), biogas and passive building, were used for the analysis. The primary research tool was a model using the concept of the so-called New Era of Innovation, emphasising the role of innovation in value creation, selection of business

processes, technical and social architecture. It allowed the characterisation and structural analysis of renewable energy sector business models, and identification of the sources and essence of the value created by those models. To add depth to the research, elements of the "Canvas" [15] business model were used, strongly identifying the elements of value creation and its delivery to the customer, of significant importance in the development of business models for new undertakings, including start-ups.

The discussion of RES business model antecedence is not limited to their identification and characterisation, but also covers the concepts and structure of the emerging business models. There is a research gap in identifying the relevant antecedents and their impact on the concepts and structural constituents of the presented RES business models.

## 2. Business Model Concepts and Their Application in the Energy Sector—Review

### 2.1. Business Models—Theoretical Background

Various definitions and concepts have amassed in the research on business models that are important for the theory and practice of management [16–21]. Due to the multitude of concepts in this area, there are proposals to develop a common research platform for a better and more effective examination of business models [19]. Taxonomical difficulties (definitions, structure, links) result from the multifaceted and interdisciplinary nature of the business model as a concept. Therefore, the related research looks at the matter from different angles, to a varying extent and depth. Theoretical and cognitive research on business models [22] typically follows either of three perspectives (research areas and aspects). These are:

- The perspective of choices made by the undertaking (strategic aspect). The choice perspective treats the business model as a dynamic, mutual interaction of strategic choices made by the organisation and their consequences, focusing on the mechanism of value creation and transfer.
- The perspective of the system of actions taken by the organisation and relevant for its activities (organisational aspect). The business model is seen as the configuration of resources and mutually dependent actions, which extend beyond the boundaries of the enterprise and so extend its operations. Value creation is emphasised here. In this perspective, the business model is seen as a dynamic structure, and its development is supported by the enterprise resources theory, the configuration theory, the organisation theory and the value chain theory.
- The normative perspective (normative aspect). From this angle, the elements that are building blocks of the business model are analysed and identified, including how they should be organised in a situation where value creation for the customer and its transfer required for profitability and organisational development underpins business operations. Here, the business model is understood mainly on the basis of the opportunity exploration/exploitation theory.

On the one hand, theoretical research on business models that has brought many interesting definitions and concepts [18,19,23–25] should be applied in practice, and on the other, practical experience informs the theory of business modelling. Regardless of the approach to the business model research, one of the key aspects is always the value as a category, its creation, retention, transfer and the recipients of that value. The basic model for value creation was proposed by Zott and Amit [5,19], and it identifies four sources of value. The first is the novelty of the transaction (new essence and structure of the transaction, new participants), another is lock-in by change cost optimisation (loyalty programmes, trust and adaptation) and by using the positive effects of external networks. The next source of value in the business model is efficiency, and in particular cost reduction, the economy of scale, simplicity and symmetry of information. The fourth of the sources is complementariness between products and services for customers, online and offline assets, technologies and activities. An important aspect is also the measurement of value. Here, the Balanced Scorecard can be used [26–28]. A big role of value creation by business

models is also pointed out by the work of Shafer, Smith, Linder [29], and also Demil and Lecocq [30] as well as Johnson, Christensen and Kagermann H. [31].

Creating unique value and competitive advantage is linked to innovation, the carrier of which is the business model [32–38]. Innovation, increasingly employed by businesses [39], enables the creation of value for customers (new products, new types of value, new customer service methods) and value for stakeholders, such as e.g., environmental protection, new technologies, new value chains or efficiency growth [40–42]. Innovation, efficiency, customer loyalty and complementariness [21], but also processes and resources that use innovation, can be seen as the factors in value creation in the business model. The implementation of innovation also seeks to prevent the threat of imitability [43]. Business models themselves may constitute an organisational innovation important for competitiveness [36], the examples of which are Economy 4.0 models [44,45].

The other vital dimension of the business model is the transfer of value for the business that generates income [25]. Its magnitude largely depends on the level of innovation, architecture, nature of the resources and processes that make up the competitive edge and harmonisation of resources and actions included in the business model [20,21]. A higher level of harmonisation between the elements of the system of activities and processes translates into higher generated value, which opens up opportunities for the incoming transfer of its bigger portion.

### 2.2. Business Models in the Energy Sector

In the case of business models for energy undertakings, innovative technologies for the generation of various types of energy (heat, cold, electricity), storage systems and efficient management of energy centres, play major roles in value creation. In the context of the presented research area of the energy sector, those business concepts that apply to the development or exchange of values (e.g., prosumers) and determine the role of innovation in the creation and implementation of business models are important. It is both about process innovation (e.g., new energy production technologies) and about equally important other types of innovation that enable better satisfaction of customer needs (including those of energy users), solution of their problems or facilitation of the exchange of values.

The dependencies between value for the customer and the transfer of value are linked to value networks and strategic choices, which are constituents of business models. When treating the business model as a specific combination of resources, it generates value through transactions both for the customer and for the organisation [15,17]. Stakeholders are also indicated as the recipients of value. It is of major importance in the case of the energy sector. For example, value created by renewable energy business models contributes to the attainment of energy policy objectives, for which the state is responsible, acting as the stakeholder in this case. Businesses that apply the multi-business model approach for the production and distribution of various energy carriers and provision of energy and related services are important in the energy sector.

Here, an integrated business model portfolio is often used, with interrelations between different business models managed by the head office [46]. Those business models should be complementary at the corporate level, and therefore are not hierarchical and should be designed to balance out one another, whilst strengthening their broader operating range [47]. An important role in the development of new business opportunities, and so value, is played by the interactive nature of the model, translating into the capacity to combine, integrate and employ both internal resources and the ecosystem [48]. In business modelling, there is also the concept of value creation and transfer in the value network [24,29,49].

The need to create innovative business models results from the fact that it is not only products that become obsolete and unacceptable for customers, but the same also applies to management processes and systems. In consequence, the existing business models are no longer capable of creating sufficient value, or the value they create may even be in decline. The innovative business model is generally referred to as a process intended to reduce costs,

optimise processes, enable access to new markets or improve financial performance [50]. In comparison to the application of product, process or marketing innovation, the innovative business model comprises a systemic change, which translates into an increase in business value, through the value proposition and the method of value generation for the customer. If the innovative business model is implemented successfully, the innovation process should result in positive transformation and renewal of the business [51]. The growth of an innovative business model typically comes from a stimulus being a change in the competitive environment, such as the pressure caused by the launch of a new, watershed technology [52]. That transition is supported by the choices of decision makers, either from the business or from politics, who monitor the evolution of the business model as used by the enterprise in order to understand how to stimulate innovation in a given sector [53]. An example of a watershed technology in the energy sector is nuclear energy or renewable energy generation, both of which have contributed to the development of new business models. The concepts and structure of innovative energy sector business models indicate that there is a need for complementariness and harmonisation of process and marketing innovation [54]. Research on new concepts for the operation of innovative energy sector businesses, relevant for the liberalising energy market, was undertaken by B. Matusiak [55]. She argues that the business model should provide the following information [55]:

- What is the specific value proposition for the model's participants?
- What consumer relationship model was used?
- What is the model for the value return on the capital employed?
- What is the configuration of all tangible and intangible assets committed to achieve that value?

In this context, B. Matusiak characterises five business models relevant for the integrated energy market [55]:

- the prosumer business model (customer–supplier exchange of value),
- the ESCO (*energy saving company*) model for the provision of comprehensive services in the area of energy management,
- the market aggregator business model (the undertaking operating between distributed energy producers and users),
- the business model of electric vehicle users in the energy market,
- the producer business model (large-scale application of sustainable, and often renewable energy sources).

The business models proposed by B. Matusiak were partially used in the development of the research concept covered by this paper, in particular as regards the application of the ESCO model.

## 3. Theoretical Research Concept

To answer the questions posited above, elements of the business model based on the principles of the New Era of Innovation were used. The operating business model according to the principles of the New Era of Innovation is presented in Figure 1. The business model is composed of three primary constituents. The first two are social architecture and technical architecture. They represent specific resources. The third constituent is business processes.

In comparing the proposed models based on the five business models of B. Matusiak [55] in the context of RES development and its constituents according to the New Era of Innovation, the authors focused on the changes that energy undertakings will undergo (Table 1).

Table 2 presents parameters of the business model used by an ESCO-type business (*energy saving company or energy service company*). ESCO companies provide comprehensive energy management services (in the area of reduction in energy consumption and demand from their customers, that is, energy users) based on delivery contracts, and provide a savings guarantee. ESCO services may include not only projects that improve the efficiency of energy consumption, but also cover equipment maintenance and repair, co-generation of

electricity and heat, new technologies, and alternative energy generation, if those services are paid for with funds from the generated savings.

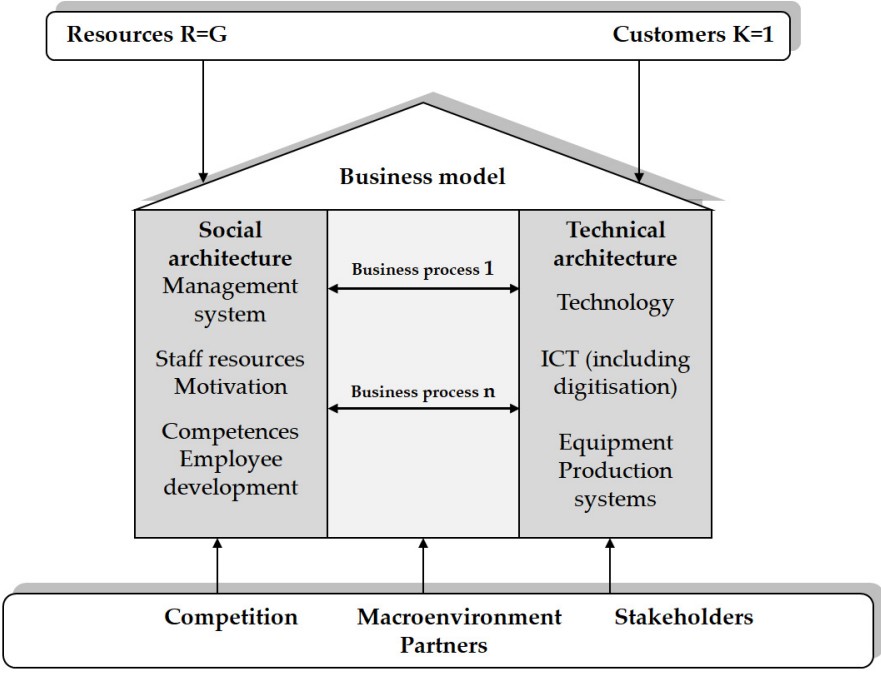

**Figure 1.** Business model concept according to the principles of the New Era of Innovation.

**Table 1.** Properties of changes that energy undertakings will undergo.

| Elements of the Business Model for the | Model | | |
|---|---|---|---|
| | End User/Prosumer | ESCO Third Party | Large Market Producers and Aggregators |
| Technical architecture | Small installations from 4 kW to 8 kW, sometimes larger. Modules from 150 W to 300 W installed. Large choice of technologies. Desired development of automatic engineering and control of modern, smart houses and new ICT (information and communication technologies). | Large renewable energy installations, calculated in terms of return on investment according to the service provider model and the energy efficiency provider model. Supporting development of ICT technologies, etc. | Large installation, PV farms and DG (*dispersed generation*) initiatives on account of taking over ESCO role, mixed models, dedicated also to small prosumers (roof lease, etc.), Aggregators - new service on the market - aggregation of energy sources and distribution, EV (*electric vehicles*) and local balancing, development of controlled demand for electricity. Development of energy storages, ICT supporting technologies, etc. necessary. |
| Social architecture | Strategic competencies include knowledge of the law on RES (*renewable energy sources*) and DSO (*distribution system operator*) obligations as regards the connection of and cooperation with prosumers, technical competence in RES installation and operation, marketing competences of the individual users market (households). | Competence in the areas of: audit of large energy facilities, project and RES investment management, energy law, energy management, investment financing, risk management. Marketing competence of the large energy user market. | Competence in the areas of: project and RES investment management, energy law, energy management, investment financing, risk management, marketing competence in the public sector and individual users market. |

**Table 1.** *Cont.*

| Elements of the Business Model for the | Model | | |
|---|---|---|---|
| | End User/Prosumer | ESCO Third Party | Large Market Producers and Aggregators |
| Business processes | Small-output RES consultancy and installation service. Energy management service to improve energy efficiency. | Energy audit, large-facility energy management, RES project management and fundraising. | RES project management and fundraising. Aggregation of public transport fleets (users) with energy distribution systems (suppliers). |
| Participants and values | End users are encouraged to invest in RES but are often sceptical and uncertain about the available technologies. | Major companies supporting the financing, special-purpose vehicles, innovation intermediaries. These are not network companies, but separate special-purpose companies. | Notably large market participants, public sector, integrated market undertakings. |

**Table 2.** Characteristics of ESCO business model parameters for RES investments.

| Elements of the Business Model | Market Participants | Brief Description |
|---|---|---|
| Primary financing | Third party—all institutions providing financing for RES projects and for the upgrade of internal infrastructure for facilities, banks (e.g., BOŚ). | Banking products, lease, loans, etc., grants and tax credits for PV investments, non-returnable funding for upgrade projects that reduce the value of the construction loans in the case of PV investments. Similar support for solar collectors, etc. |
| Support financing | White certificates | Trading in white certificates if significant energy efficiencies are achieved in the projects implemented by ESCO |
| Advanced financing (e.g., micro-network development) | Private funds, venture capital, special-purpose foundations, special-purpose vehicles, lease, etc. | Such financing requires long-term and stable tax policy to apply credits and concessions. This applies to e.g., wholesale gas prices, liberalised for trade negotiations, or liberalised prices for small users. |
| Technology partner | ESCO and, through ESCO, companies supplying panels and installations and wind power equipment along with installation and support (ESCO responsibility), additional required equipment and on-site installations. | Market offering is very broad and varied in terms of price, the selection of the cell technology is of major importance for its efficiency and costs. |
| Technology partner | DSO and/or market aggregator | Connection agreement and power at termination point, installation of meters and metering systems for which DSO is responsible. Potentially cooperation with the market aggregator for the balancing of ESCO customers (in the case of full energy market integration). |
| In the case of electricity and other utilities, purchase of power from the network for own needs | The selected electricity seller, according to the TPA principle (or the aggregator taking over the billing services) | Billing and balancing on a contract basis for a large business customer (ESCO); based on free-market prices. |
| In exceptional cases, sale of energy to the network, if stipulated by the contract, e.g., lease of customer's roof - to generate energy for the network (producer model) | State regulation implemented by the official seller of electricity. | Regulation under the RES Act: installation enabling the generation of green certificates. |

Source: [54].

The implementation cost of energy-efficient projects is paid by the ESCO company that, over the term of the contract, participates in the proceeds from the investment or upgrade. In other words, the investor repays the cost of the upgrade project using operating savings generated.

The ESCO business proceeds to execution only when the return on capital employed in the project is sufficient. If the flow of funds for the ESCO business from the energy savings over the life of the contract is smaller than the combined cost, the ESCO company runs at a loss. In summary, it may be argued that value based on innovation is the central dimension of the contemporary business model. The business model as such is a concept describing the logic behind the activities of a business capable of creating value for its customers and stakeholders, the sources of which are the various innovations, which clearly relates to the energy sector.

On the basis of theoretical business modelling studies and our own research, the business model for the prosumer energy generation sector is defined as a configuration of business processes that combine and develop resources in the form of social and technical architecture, creating value based on renewable energy sources. The value is created for the customer, but may also be created by the customer.

According to our work, the business model can be treated as a system for the configuration of resources and interdependent activities focused on value creation. The collection of such activities, resources, the method of their organisation and links between activities, resources and the value network, which enable such activities in cooperation with partners or customers, are all clearly dependent on the adopted business model [15,19,32]. Many discussions emphasise close links between the business model and value creation for customers and the business, referring to the role of configuration of tangible and intangible factors and the option to transfer some of the income from the value provided to customers [25,33,34]. The value translates into the profitability and competitive edge of the business [17,25,35]. Two main dimensions are prominent in the business model concepts. The first is how the value for the customer is created, and in particular, what are its characteristic features that determine customer satisfaction. Properties of value created by contemporary business models are given in Table 3.

**Table 3.** Properties of value created for customer by business models.

| No. | Type of Value for the Customer | Property Characterising the Value |
|---|---|---|
| 1 | **Emotional values** | Beauty, pleasure, willingness, love, comfort, emotional bonds, sentiment, interest, scale of experience and impression ("adrenaline lovers"), pride, satisfaction |
| 2 | **Technical values** | Strength, performance, ergonomics, innovation, usefulness, functional fit, easy operation, reliability, lightweight structure, energy efficiency, shape (design), other specific quality properties |
| 3 | **Economic values** | Attractive price, availability, delivery time, convenient terms of payment, timing, broad selection of offered productions, logistics links in services (e.g., the existence of the repairer network for electric cars) |
| 4 | **Social and ethical values** | Quality of life (healthy food, sports and recreation), health, ability to pursue one's interests and hobbies, product liability of the business, working conditions, fair compensation, fair trade, opposition to globalisation processes and the domination of global corporations (proponents of "free software", etc.), opposition to planned obsolescence (programmed product aging), safety |
| 5 | **Organisational values** | Prestige, organisational efficiency, coordination, image, quality of domain leadership (e.g., recognition by brand and visionary and charismatic leadership, e.g., S. Jobs—Apple) |
| 6 | **Environmental values** | Product composition, production method (e.g., RES), disposability, option to be processed and reused, environmental impact (RES), long-term environmental impact (e.g., proponents of ecological farming), capacity not to generate the carbon footprint. |

What should be noted here is the growing variety among those properties, reaching far beyond the boundaries set by such traditional values as price, quality, availability or functional suitability of the product. Changes in awareness, lifestyles, and customers' access to knowledge produce ever newer needs and preferences. This is where various innovations in business models can be broadly applied. From the standpoint of the analysis and structure of business models, it is important to know what elements of the business model play a major role in value creation and how that value will be realised [15,17].

## 4. Research Methods and Materials

Our own research covers the discussion of key antecedents (external and internal factors) impacting business models applied by RES undertakings. The method used is desk research, that is, review of data from available sources, covering in particular compilation, cross-verification and processing. The review formed the basis for conclusions that address the following question: What are the most important antecedents for the development of innovative business models in the RES-based energy sector?

Subsequently, case studies were employed to analyse the selected business models and assess their value. Here, qualitative research methods were applied to understand the uniqueness of the case, nature of a specific phenomenon, its context and interaction with other elements, rather than to anticipate what may happen in the future [56]. Regardless of criticism of case studies as a scientific method, it can be argued that they follow methodological rigour requirements, and the results of case studies lead to important conclusions that verify, update and amplify knowledge generated by science [56]. The primary intent of qualitative studies, including case studies, is to explore the circumstances of phenomena, their reasons and dependencies, and often also to come up with a new theory. If properly applied, the case study method is rigorous, complex, solid and provides strong practical evidence [57].

The research process chart is shown in Figure 2.

The first step was analytical and background research on the antecedence of business models and business model theories, in particular, their definitions, structure and value creation aspects.

In the second step, a new research concept was presented to analyse elements of business models applied by RES companies, with a particular focus on the ESCO model. This enabled the identification of the value proposition offered by those models.

The third step comprised:

- analysis of external factors (energy policy, legal and regulatory aspects, energy markets, macro-economic situation, research and development, climate policy, sustainable development, responsible business),
- analysis of internal factors (situation of the sector, ownership structure, electricity generation mix, renewable energy generation financing and support, innovative power sector technologies, Economy 4.0 instruments).

The basis for the desk research was public statistics documents, reports, studies, publications and statistical yearbooks.

The antecedence of RES business models is not limited to their identification and characterisation, which is presented in step three, but also covers the analysis of the concepts and structure of the emerging models (step two). The proposed multi-case study covered the identification of:

- Elements of social architecture, i.e., human capital (including prosumers), strategic competences and skills, knowledge resources;
- Items of technical architecture, covering their own tangible resources, sources of supply, organisation and ICT systems;
- Business processes.

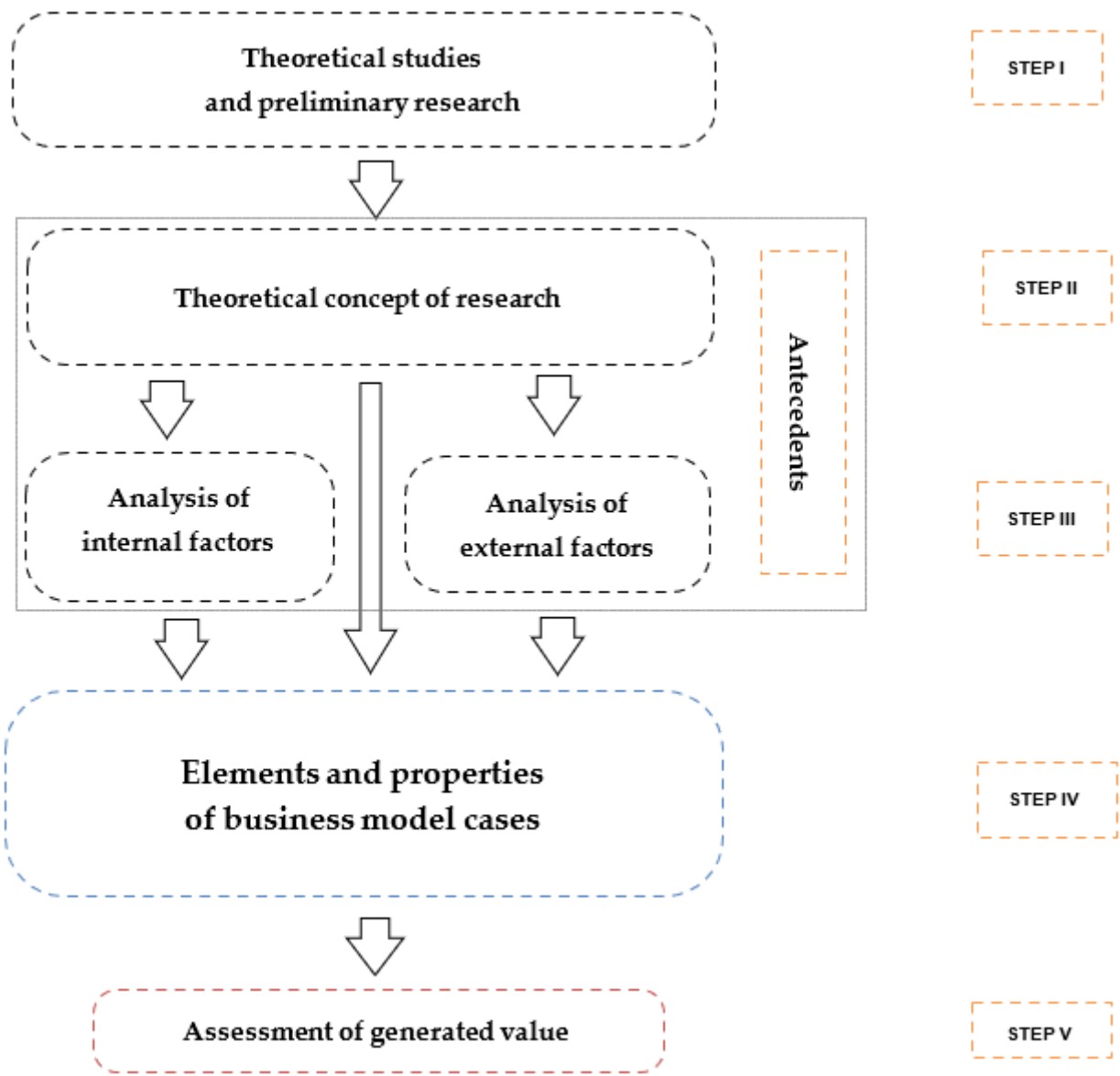

**Figure 2.** Research methodology scheme.

Research method triangulation was used for the case studies. Non-participant observation, unstructured interview and business documentation review were used for the research.

In the last step, the value created by the business model was assessed in terms of its sources and measurable effects. The social aspect of the created value is of importance in the case of innovative energy generation projects.

## 5. Findings and Discussion

### 5.1. External Factors

One of the important aspects of business model development for innovative energy generation is the need for and nature of RES. Renewable energy is energy generated from natural, recurring processes. Renewable energy sources represent an alternative to conventional, fossil and non-renewable energy carriers (fossil fuels). In Poland, energy from renewable sources covers energy from direct solar radiation (transformed into heat or electricity), wind, geothermal resources, water, solid biomass, biogas and liquid biofuels. Renewable energy sources share the following properties [58]:

- RES resources are continuously replenished by natural processes,
- they can be managed in a way that ensures that they are never exhausted,
- their use does not result in the emission of pollutants (or other substances), which means they have a relatively small environmental impact,
- equipment for the production and distribution of renewable energy is often innovative in nature.

An important trigger for changes in innovative energy generation models (including in Poland) is the EU energy policy [59,60], implemented under national policy documents [45]. The first meetings on environmental protection were held in 1968 at the UNESCO Inter-governmental Conference of Experts devoted to the interrelations between the natural environment and growth, at which an international and interdisciplinary "Man and the Biosphere Programme" [61] was developed. For many years, the EU has been the leader of environment-friendly energy generation. Poland, by acceding to the European Union, in addition to many other privileges and benefits, also has the duty to fulfil certain obligations as a member state. One of them is protection of the natural environment by an increasing use of RES. The extent of renewable energy use in EU member states is governed by EU documents and normative acts that set forth general and particular targets of certain shares of renewable energy in the overall consumption of that energy [62]. As part of its environmental obligations, the EU set quantitative targets for 2020 in the "3 × 20%" energy and climate package, that is, a reduction in greenhouse gas emissions by 20% relative to 1990, a reduction in energy consumption by 20% in comparison to EU forecasts for 2020, and an increase in the share of renewable energy sources by 20% of total consumption in the EU, including in the use of renewable energy sources in transport by up to 10% [63]. In the light of those obligations, changes are necessary in Polish energy generation, which also entails changes to business models. In 2001, the European Parliament adopted Directive 2001/77/EC on the promotion of electricity produced from renewable energy sources in the internal electricity market. The directive specified the share of electricity from renewable energy sources in the total energy consumption for all EU member states at the level of 22% in 2010. This was the first document devoted entirely to supporting RES [64]. The year 2003 saw Directive 2003/30/EC on the promotion of the use of biofuels or other renewable fuels for transport. The purpose of the directive was to promote the use of biofuels or other renewable fuels instead of diesel oil or gasoline [65]. Another important legal document for the growth of renewable energy sources was the 2007 Renewable Energy Road Map - Renewable energies in the 21st century: building a more sustainable future. It provides a long-term perspective on renewable energy sources in the EU. The road map offers a goal of a 20% share of renewable energies in total energy consumption and a 10% share of renewable energy in transport by 2020. The specification of targets at the European level will ensure a degree of stability in terms of national policy in this respect [66]. Particular commitment of EU member states to increase their share of RES production and consumption results from the obligations of the EU climate and energy package approved in 2008. The document requires member states to cut $CO_2$ emissions by 20% relative to 1990, increase the share of renewable energy in the total energy consumption in the EU to 20% and increase energy efficiency by 20% [67]. In December 2019, the European Union adopted a comprehensive strategy on environment protection and climate change, according to which Europe is to become climate-neutral by 2050. This strategy is the European Green Deal, which is to help transform the EU into a modern, energy-efficient and competitive economy:

- which, in 2050, will achieve zero net greenhouse gas emissions,
- in which economic growth will be decoupled from resource use,
- in which no person and no place will be left behind.

In Poland, there are divergent views on the climate and energy package from the standpoint of the Polish economy and interests of the Polish businesses, including the coal mining sector. The package stipulates that EU member states will publish reports every five years describing their actions to achieve the targets and, every two years, reports

on the results achieved. The reports are evaluated and commented on by the European Commission. Legal tools established and used by the EU to support a greater share of renewable energies are varied in nature and range of impact. Key documents governing the use of RES in Poland include the 1997 Energy Law, which not only defines renewable sources, but also specifies the conditions for sustainable development of the country [68]. The Energy Law provided the legal basis for increasing the use of renewable energies by requiring energy undertakings to permit applicant entities to connect to the grid. The support for projects intended to reduce the consumption and loss of energy supplied to users and the replacement of conventional energy sources with non-conventional sources is governed by the Act of 21 November 2008 on the support for thermal insulation and refurbishment. The Polish legal framework governing the generation of energy from renewable sources is closely linked to the acts of EU law. The most important document that governs RES is the Act of 20 February 2015 on renewable energy sources, which sets forth [69]:

- rules and conditions for the generation of electricity from renewable energy sources, agricultural biogas in renewable energy installations, biofuels.
- mechanisms and instruments supporting electricity generation from renewable energy sources, agricultural biogas, heat from renewable energy installations.
- rules of issuance of guarantees of origin for electricity generated by renewable energy sources in renewable energy installations.
- rules for the execution of the national action plan on energy from renewable sources.
- conditions and procedure for the certification of contractors for micro-installations, small and renewable energy installations whose total installed heat capacity is not more than 600 kW, and training organiser accreditation.
- rules of international cooperation on renewable energy sources and joint investment projects. The document also contains the definition of the energy cluster as follows: An energy cluster–civillaw agreement which may include natural persons, legal persons, scientific and research institutions and local government entities, concerning the generation and balancing of demand, distribution or trading in energy from renewable energy sources or other sources or fuels, in a distribution network whose rated voltage is less than 110 kV, where the operating area of the cluster is not greater than a single district within the meaning of the Act on the district-level government, or 5 communes within the meaning of the Act on the commune-level government; the cluster is represented by the coordinator, being a cooperative established for the purpose, association, foundation or any member of the energy cluster indicated in the civil law agreement.

Poland's situation against the EU's background provides a stimulus for the development of innovative (renewable) energy generation business models. The share of renewable energies in the Polish energy mix has recently increased. The situation is much better in the European Union as a whole, where renewable energies have been used on a much larger scale and for much longer, unlike in Poland, which is still developing in this respect [70] (Table 4).

**Table 4.** Share of energy from renewable sources (in % of the total gross energy consumption).

| Years | 2012 | 2013 | 2014 | 2015 | 2016 | 2017 | 2018 | 2020—Objective |
|---|---|---|---|---|---|---|---|---|
| **UE-28** | **14.4** | **15.2** | **16.1** | **16.7** | **17.0** | **17.5** | **18.0** | **20** |
| **Austria** | 31.4 | 3 | 2.3 | 33.5 | **33.4** | 33.1 | 33.4 | **33.0** |
| **Czech Republic** | 12.8 | 13.8 | 15.1 | 15.1 | **14.9** | 14.8 | 15.1 | **13** |
| **Finland** | 34.4 | 36.7 | 38.7 | 39.3 | **39.0** | 40.9 | 41.2 | **38** |
| **France** | 13.4 | 14.1 | 14.7 | 15.0 | **15.7** | 16.0 | 16.6 | **23** |
| **Netherlands** | 4.7 | 4.8 | 5.5 | 5.7 | **5.8** | 6.5 | 7.4 | **14** |
| **Lithuania** | 21.4 | 22.7 | 23.6 | 25.8 | **25.6** | 26.0 | 24.4 | **23** |
| **Germany** | 12.1 | 12.4 | 13.8 | 14.9 | **14.9** | 15.5 | 16.5 | **18** |

**Table 4.** *Cont.*

| Years | 2012 | 2013 | 2014 | 2015 | 2016 | 2017 | 2018 | 2020—Objective |
|---|---|---|---|---|---|---|---|---|
| **Poland** | **10.9** | **11.4** | **11.5** | **11.9** | **11.4** | **11.1** | **11.5** | **15** |
| **Slovakia** | 10.4 | 10.1 | 11.7 | 12.9 | **12.0** | 11.5 | 11.9 | **14** |
| **Italy** | 15.4 | 16.7 | 17.1 | 17.5 | **17.4** | 18.3 | 17.8 | **17** |
| **Denmark** | 25.7 | 27.4 | 29.3 | 30.8 | **33.3** | 35,8,0 | 36.1 | **30** |
| **Norway** | 100.0 | 100.0 | 100.0 | 100.0 | 100.0 | 100.0 | 100.0 | - |
| **Iceland** | 95.4 | 96.7 | 97.1 | 93.1 | **95.3** | 95.1 | 96.2 | - |

Source: Erneuerbare Energien in der EU Weiterer Anstieg des Anteils erneuerbarer Energien am Energieverbrauch in der EU auf fast 17% im Jahr 2015, nr 43/2017 http://ec.europa.eu/eurostatEnergy from renewable sources in 2019, Statistics Poland, Warsaw 2020 Source: Eurostat (nrg_100a) and (nrg_107a) Eurostat (nrg_ind_335a).

### 5.2. Internal Factors

In terms of business model development, RES growth tendencies relevant for each type of such installations are important. Table 5 presents the installed capacity in various types of renewable energies in Poland in 2012–2020. The figures indicate a dynamic, even though non-steady, growth of renewable energy generation in Poland. The growth rates vary with the type of RES installations. The highest growth can be seen in PV and wind generation, while the growth in hydroengineering is sluggish. Currently (2021), the total installed RES capacity in Poland was about 15.3 GW, while the total domestic capacity was 53.3 GW, which represents an increase in RES in annual terms by 31.3% (rynekelektryczny pl portal, PSE portal) Most of the installed RES capacity comprised:

- photovoltaic plants (including micro-installations)—6126 GW,
- wind power plants—6828 GW,
- biogas plants—0.348 GW,
- biomass plants—0.997 GW,
- hydropower plants—1.022 GW.

**Table 5.** The installed capacity of the renewable energy generation sector in Poland in 2012–2020 by type of energy installations that hold a licence or are entered in the register of energy sector activities.

| Type of RES Installation | 2012 | 2013 | 2014 | 2015 | 2016 | 2017 | 2018 | 2019 | 2020 |
|---|---|---|---|---|---|---|---|---|---|
| | **Installed Capacity (MW)** | | | | | | | | |
| Biogas installations | 131.3 | 162.2 | 188.6 | 212.5 | 234.0 | 235.4 | 237.6 | 245.4 | 255,4 |
| Biomass installations | 820.7 | 986.9 | 1008.3 | 1122.7 | 1281.1 | 1362.0 | 1362.8 | 1492.9 | 1512.9 |
| PV installations | 1.3 | 1.9 | 21.0 | 71.0 | 99.1 | 103.9 | 147.0 | **477.8** | **887.4** |
| Wind power installations | 2.5 | 3389.5 | 3833.9 | 4582.0 | 5807.4 | 5848.7 | 5864.4 | 5917.2 | 6347.1 |
| Hydropower installations | 966.1 | 970.1 | 977.0 | 981.8 | 994.0 | 988.4 | 981.5 | 973.1 | 976.1 |
| **Total** | **4416.0** | **5510.6** | **6028.6** | **6970.0** | **8415.5** | **8538.3** | **8593.4** | **9106.3** | **9979.2** |
| Growth y/y | 1334.1 | 1094.6 | 518.0 | 941.4 | 1455.5 | 122.8 | 55.1 | 512.8 | 872.9 |

Source: Energy Regulatory Office 2020 ure@ure.gov.pl; www.ure.gov.pl.

The presented data indicate a high growth rate and the need to intensify RES investment activities resulting from the energy policy (both national and EU) targets. This entails a broader than ever application of innovative energy generation business models. The above is confirmed by Poland's Energy Policy by 2040, adopted in 2021, which responds to key challenges ahead of the Polish energy generation in the decades to come, and charts the directions of energy sector development, considering tasks that need to be implemented in the short- and medium-term, offering strategic choices in terms of technologies to build a low-emission energy system. The objectives of the adopted energy policy include energy security whilst ensuring competitiveness of the economy and improvement in its energy efficiency, along with the reduction of the energy sector's impact on the environment, while making optimum use of the country's own energy resources. Each of the particular objectives of PEP 2040 and all actions and strategic projects it envisages are encompassed by the above three aspects of the objective. The policy focuses on upgrade and innovation,

seeks to stimulate economic growth, efficiency and competitiveness, and promotes care for the environment and climate. Energy transition will be based on the following three pillars:

- Pillar One: Fair transition,
- Pillar Two: Zero-emission energy system,
- Pillar Three: Good air quality [71,72].

The growth in the importance of renewable energy sources results from the need for a low-emission energy transition through the diversification of the energy balance and reduction of its emission rates, contributing to the EU-wide 32% target for RES in total gross energy consumption, and the falling costs of such energy generation technologies. According to studies under the European Green Deal, over 75% of greenhouse gas emissions in the EU come from energy production and use. Decarbonisation of the EU energy system is therefore of paramount importance for the attainment of our climate objectives for 2030, and implementation of the EU's long-term strategy for carbon emission-neutrality by 2050. The European Green Deal focuses on three primary assumptions regarding the transition to clean energy, which will help to reduce greenhouse gas emissions and improve the quality of our lives. Those assumptions are:

- ensuring price-competitive and safe energy supply in the EU,
- development of a fully integrated, interconnected and digital EU energy market,
- prioritising energy efficiency, improvement of the energy characteristics of buildings, and development of the energy sector based primarily on renewable sources.

In effect, the RES target assumes that solutions will be implemented that will enable a reduction in $CO_2$ emissions and meet growing needs. The plan estimates that, by 2050, over 80% of electricity will come from RES and will cater for one half of the final energy demand in the EU. To meet the growing demand, the current production level needs to increase nearly two-fold. The implementation of renewable energy generation solutions allows the decarbonisation of such sectors as system heating, transport and industry. Poland has declared that it will achieve a RES share of at least 23% in total gross energy consumption in 2030. Considering the expected technology development, a special role in the pursuit of the RES target will be played by offshore wind farms, the growth of which is a strategic decision concerning the development of key competences in this respect, enabling Poland's economic growth. Continued growth of PV energy generation is projected, whose generation profile matches energy demand peaks in summer, as well as of onshore wind farms, which generate electricity in similar time intervals to offshore wind generation. A significant increase in the importance of biomass, biogas and biofuels is envisaged. In addition, RES-based distributed energy generation growth is projected, as well as sale, storage or participation in DSR programmes by individual entities (e.g., active users, renewable energy and other prosumers) and energy communities (e.g., energy clusters, energy cooperatives). By 2030, an approximately five-fold increase in the number of prosumers is projected, and an increase to 300 in the number of areas demonstrating energy sustainability at the local level.

The energy sector in Poland, similarly to other sectors in 2020, was marred by the effects of the COVID-19 pandemic. Key factors in the sector in the last year included:

- the share of coal in electricity production dropped below 70% in 2020.
- an increase in the RES share owing to prosumer PV systems and onshore wind power plants. The number of gas-fired cogeneration units increased.
- PV installed capacity grew to 4 GW.
- electricity production dropped by 3.8%, and the corresponding demand dropped by 2.1%.
- net imports of electricity reached an all-time high of 13.3 TWh, which accounts for 7.8% of electricity consumption in Poland.

The consumption and production of energy coal dropped. Coal stock increased, totalling 15 mio. tonnes at the end of 2020. In 2020, the installed capacity of the National Power System was 49,238 MW, and available capacity was 49,095 MW. Commercial power

plants accounted for 36,364 MW of the installed capacity, including 22,747 MW in hard coal-fired plants, 8478 MW in brown coal-fired plants, holding, respectively, shares of 46% and 17% in the system. RES (notably wind power) installed capacity totalled some 10,229 MW, while industrial power plants (various fuels) accounted for 2645 MW. Over 95% (23.8 GW) of all controllable sources of power in the National Power System are owned by companies with the Polish Treasury's shareholding. The current generation structure primarily results from historical considerations and energy resources in Poland's possession. The Polish electricity generation mix has not changed significantly over the years. The system uses mainly conventional fuels, such as hard coal and lignite. However, the share of those fuels in the total energy mix was nearly 70% in 2020. The electricity generation sector in Poland relies on centrally dispatched conventional generation units that produce electricity from coal. Over the last decade, however, we have seen a major increase in wind power and other renewable energy sources. In 2020, the installed capacity in the sector grew from 7490 MW to 10,229 MW. The growth of RES sources of electricity, including for electricity generated by prosumers and energy clusters, contributes to a gradual transition from a passive (one-way) to an active (two-way) network.

The position of the respective power sector entities has not changed significantly in recent years. The three largest producers are members of corporate groups with the shareholding of the Polish Treasury, such as e.g., PGE Polska Grupa Energetyczna S.A., and ENEA S.A., TAURON Polska Energia S.A., which hold nearly two-thirds of the installed capacity, generating some 67% of the electricity in the country.

Determinants of sustainable development, dynamic economic and social changes in the contemporary world are closely related to the search for and use of renewable energy sources (RES) that are safe for the natural environment. RES investments can be developed by almost all energy markets participants, that is, energy producers (corporations and other energy undertakings, business and non-business organisations, individual customers). A vital role in RES development is played by municipalities, which are often the administrators of funding for that purpose. The funding comes from internal or third-party funds (including EU state aid funds) for residents or SMEs. Communes may act as project owners or co-owners for RES projects, or establish favourable conditions for investors willing to commit funds to those energy sources. Energy generation from renewable sources primarily depends on local or regional considerations, and therefore the related opportunities should be discussed separately for each area [73]. There is a significant potential in Poland for the development of innovative, local, distributed energy generation. A positive approach to renewable energy sources by local government officials is also clearly seen here [74].

*5.3. Case Studies of Chosen Business Models*

The identified antecedents impact and inform business model concepts and structures for the renewable energy sector. To confirm their impact, examples of business cases of selected energy undertakings have been analysed.

Using the discussed research methodology, the following business models are presented in the following sections:

- photovoltaic farm,
- passive building,
- local (communal) biogas plant.

The models account for the current status of the Energy Law and energy policy as it relates to RES, and the examined projects represent selected RES applications in the business and public domain.

The basis for the description of each case was the model of the New Era of Innovation which, in combination with the value creation process, offered options for the presentation of the model architecture and business processes, as well as sources and effects of value creation.

5.3.1. Photovoltaic Farm in a Project Owner-Managed Undertaking

The business model for this type of the undertaking involves the use of technology transforming sunlight energy into electricity, in other words, the generation of electric current from solar radiation using the photovoltaic effect. Elements of this business model are presented in Table 6.

**Table 6.** Elements and features of the business model of a photovoltaic farm developed by a project owner.

| Elements of the Model | Life Span — 20 Years (Launch on 1 March 2019) |
|---|---|
| **Social architecture** | |
| Human resources (including prosumers) Strategic competences and skills and knowledge resources | - Installation technician (contract of mandate), tenants (owners of flats in an estate). Farm management by natural persons and recurring contract with the maintenance technician.<br>- Strategic competencies include knowledge of the law on RES and DSO duties in respect of connection and collaboration with prosumers. Bridge funding acquisition skills with the project owner. Knowledge of net energy billing through balancing. |
| **Technical architecture** | |
| Tangible resources (size and structure of the assets, characteristics of the potential, sources of supply, organisation, global resources) ICT resources (systems) | - holding an apartment under the ownership title<br>- RES installation of 960 units of Sharp Solar 275 Wp<br>1. efficiency of ca. 98%<br>- connection as well as metering and billing system<br>- application of smart networks and the necessary software, including a two-way meter.<br>- use of the internet and smartphones for net metering by flat owners.<br>- smart grid, smart metering |
| **Business processes** | |
| Process map. Business processes | The primary business process is the generation of electricity from solar energy. The remaining processes are the optimisation of own energy consumption, energy billing and balancing. Net-metering and net energy billings means that the DSO deducts from the bill the produced excess supplied to the network, charging only the transmission charge for the energy supplied by DSO. Potential process control over the internet or smartphone. Maintenance and repair process (after the warranty term). An important business process is also energy management (balancing, billing, consumption optimisation). |
| **Value created** | |
| Sources of the value | The primary source of the value is the application of ES. Value for the prosumer is the electricity produced that is used for own needs and its excess is transferred to the energy undertaking for balancing. Ensuring energy self-sufficiency (in terms of electricity). Macro-scale growth in energy security. |
| Effects | - energy production—230,400 kW/h.<br>- savings on own energy production—EUR 31,322 (PLN 149,760).<br>- fixed costs of energy distribution—EUR 3855 (PLN 18,432).<br>- total savings resulting from the electricity charges: EUR 27,467 (PLN 131,369) per year on average. |

The use of smart networks and smart metering software enables the transfer of excess energy to the network. The project is part of the residential development project "Sunny hill" in Katowice, comprising 24 buildings, with four flats in each, in the first stage of the project. The installation will be operated by residents of the estate and the farm's maintenance technician. The installed capacity of each building is 11 kW. The presented PV farm model comprises certain constituents of the ESCO model (RES energy generation, energy management in terms of balancing and billing, and optimisation of consumption for energy generated by in-house sources to enable its saving). The direction in which

this model is likely to develop is the installation of heat pumps to offer new value to the residents, providing heat from in-house sources to improve energy security.

### 5.3.2. Low-Emission (Passive) Office Building in the Science and Technology Park Euro-Centrum in Katowice

Passive buildings demonstrate a markedly lower demand for energy than conventional buildings. In effect, the costs of heating and electricity supply are reduced. The application of alternative energy sources in such buildings avoids the cost of environmental pollution from low-stack emission sources. Passive construction is promoted across Europe. An example of that solution is a low-emission (passive) office building at the Science and Technology Park Euro-Centrum. The building has five floors and floor space of 8100 m$^2$, with the social and office space occupying 5500 m$^2$, and the rest being used for laboratories. Elements of the business model for this project are shown in Table 7.

**Table 7.** Elements and properties of the low-emission office building.

| Elements of the Model \ Life Span | At Least 30 Years (Launch in February 2014) |
|---|---|
| **Social architecture** ||
| Human resources (including prosumers) Strategic competences and skills and knowledge resources | - Employment of experts in the area of energy generation and environmental protection. <br> - Skills of supporting the systems and installations applied. Skills of managing a large facility. |
| **Technical architecture** ||
| Tangible resources (size and structure of the assets, characteristics of the potential, sources of supply, organisation, global resources) ICT resources (systems) | Structural solutions applied: <br> - column and slab system, <br> - wall insulation with 30 cm thick Styrofoam, <br> - glazed building centre ensuring maximum daylight penetration, <br> - triple-glazed windows with high insulation performance, with heat transfer coefficient of 0.7, <br> - smart grid, smart metering, <br> - The building installation comprises e.g.,: <br> - 6 heat pumps providing heat to the building by increasing water temperature <br> - 10 solar collectors (vacuum), 3 systems of photovoltaic panels installed on the roof (roof panels) <br><br> The building has a Data Centre that enables data collection and processing, thanks to advanced technologies and equipment. |
| **Business processes** ||
| Process map. Business processes | The primary process is building management, ensuring the integration and management of installations from a single location and control of operating parameters of the respective equipment items. The remaining processes are: <br> - energy management (balancing, billing, consumption optimisation), <br> - energy and heat generation process, <br> - building administration, <br> - lease and cooperation with tenants, <br> - maintenance and repair process, <br> - building operating staff management. |
| **Value created** ||
| Sources of the value | The primary source of value is the application of a RES solution and generation of energy savings. |
| Effects | - Value for the tenants is modern and safe interiors, good location. <br> - The capacity of the photovoltaic plant is sufficient to cover the annual demand for energy from the building's systems, such as heating, cooling and ventilation. <br> - Increase in macro-scale energy security. <br> - Development of the image and promotion of the Science and Technology Park Euro-Centrum in Katowice as a centre for renewable energy and energy efficiency. |

The Euro-Centrum passive building complies with EU requirements for such structures, both in technical and economic terms. According to the assumptions, the building consumes only 12.5% of the energy used by a conventional building. In 2012, the structure

received an award in the Silesian Innovator competition, and in 2013, a European Green Building Award. The project was funded by the European Regional Development Fund (Operational Programme Innovative Economy 5.3). The low-emission office building is a local version of the ESCO model, as it generates electricity and heat, providing them to users (residents) of the facility, as well as manages and optimises the consumption of energy, which brings energy savings. A development option for this model may be the application of energy storage that allows for a fuller utilisation of energy from the in-house RES (keeping surplus energy rather than sending it to the distribution system operator) to improve energy security.

### 5.3.3. Communal Biogas Plant Business Model

The presented model is used by a biogas plant currently under construction in a commune in southern Poland. The biogas plant is fitted with an integrated co-generation system combined with a gas turbine for heat and electricity generation. The supplied medium is agricultural biogas (from renewable sources) generated from communal substrates, which means that the energy is generated from recurring natural processes occurring in renewable energy sources. The operator of the infrastructure is a separate entity established within a municipal utilities company. A modern biogas plant, 370 kW in installed capacity, is fitted with a modular co-generation system, with electrical output of 370 kW and heat output of 410 kWt. The communal biogas plant will contribute to the diversification of heat and electricity production from renewable sources, supplying such facilities as a sewage treatment plant, a school and a kindergarten, an indoor swimming pool and a palace museum. Elements of this business model for the biogas plant are presented in Table 8. The agricultural biogas produced by the biogas plant will be generated from local substrates and will also supply, as an alternative, through an off-premises I&C system, a distributed co-generation module with 55 kWe/88 kWt in output.

**Table 8.** Elements and properties of the municipal biogas plant business model.

| Elements of the Model / Life Span | 15–20 Years (Launch on 1 March 2016) |
|---|---|
| **Social architecture** | |
| Human resources (including prosumers) Strategic competences and skills and knowledge resources | - Hiring of 5 energy sector experts. <br> - The remaining functions, i.e., administration, finance and legal, are performed by the communal utilities company. Strategic competences include: energy engineering and distribution of electricity and heat, knowledge of the law on RES and DSO duties in respect of connection and collaboration with local energy sources. Skill of raising external funds. |
| **Technical architecture** | |
| Tangible resources (size and structure of the assets, characteristics of the potential, sources of supply, organisation, global resources) ICT resources (systems) | - equipment for the storage and fermentation of substrates, integral co-generation system with a gas turbine for electricity and heat generation with 370 kWe in electrical output and 410 kWT in heat output, <br> - the I&C system enables the supply to a distributed cogeneration module with 55 kWe/88 kWt in output, <br> - ICT systems for biogas, heat and electricity production process control, <br> - smart grid, smart metering. |

**Table 8.** *Cont.*

| Elements of the Model / Life Span | 15–20 Years (Launch on 1 March 2016) |
|---|---|
| **Business processes** | |
| Process map. Business processes | The primary process is the generation of electricity and heat from biogas. The remaining processes are:<br><br>- energy management (balancing, billing, optimisation of consumption),<br>- contracting of substrate supply (grass, corn, manure, hay, fruit waste, etc.),<br>- substrate supply logistics,<br>- administrative and financial support,<br>- customer support (sewage treatment plant, school and kindergarten, indoor pool and palace museum),<br>- maintenance and repair process (after the warranty term),<br>- cooperation with a local DSO. |
| **Value created** | |
| Sources of the value | The primary source of the value is the application of ES. Value for residents of the commune is the generated electricity and heat used in public facilities. |
| Effects | - reduction in the quantity of waste deposited in the environment.<br>- air quality improvement.<br>- emission reduction by 1117 $Mg/CO_2$/year.<br>- production of 995 MWh of electricity and 682 MWh of heat.<br>- improvement in energy security and increase in energy efficiency of communal public buildings.<br>- ensuring a backup source of supply for the above communal buildings.<br>- development of the commune image and promotion of efficient energy use through the development of agricultural gas-fired co-generation systems. |

A significant advantage of the presented business model is the creation of sizeable social value. The investment helps to manage local green resources, including waste. The generated electricity and heat will supply public buildings that are primarily used by the commune's residents. The feasibility of such projects largely depends on the existing assistance funding options. In this case, the co-funding rate was over 59%. The biogas plant model implements in-house energy generation processes and those that enable energy supply to communal establishments. The model also provides for energy management in terms of balancing, billing, and optimisation of consumption. There are similarities between those processes and the ESCO model processes. The model can evolve into using energy storage to conserve energy produced in low-demand periods for auxiliary loads during peak demand periods.

At the last stage, the value created by the business model should be assessed in terms of its sources and measurable effects.

*5.4. Assessment of Value*

The types of value created by the examined models vary in nature, as shown in Table 9. Clearly, the dominant and also shared features are energy generation in an environmentally friendly way, innovation (mainly process and marketing innovation), and quality of life improvement, health in particular. A measurable effect of the latter property is an improvement in air quality, which is very important in a situation where the Polish energy generation sector relies on coal in over 80%, and smog is the cause of death of over 50,000 Polish residents. Renewable energy production also translates into savings on ever more expensive emission allowances by conventional power plants. Energy production by prosumers or communal producers offers significant savings on energy carriers, which is all the more important against the backdrop of steep rises in their prices.

**Table 9.** Values created by the selected innovative energy generation models.

| | Photovoltaic Farm | Passive Building | Biogas Plant |
|---|---|---|---|
| **Emotional values** | enjoyment, comfort, emotional bonds, interest, satisfaction | beauty, enjoyment, comfort, interest, scale of emotions, pride | willingness, sentiment, interest, pride |
| **Technical values** | innovation, lightweight structure, energy efficiency | ergonomics, innovation, usefulness, lightweight structure, energy efficiency, shape (design), other specific quality properties | innovation, suitability, usefulness, easy use, reliability |
| **Economic values** | availability, delivery time, timing, broad range of offered products | broad range of offered products | attractive price, convenient payment terms, logistics links in services |
| **Social and ethical values** | quality of life (in particular health), corporate responsibility for the product, fair compensation, fair trade, energy security | quality of life (including health), capacity to pursue interests and pastimes, fair trade, opposition to the policy of planned product obsolescence energy security | quality of life, fair trade, energy security |
| **Organisational values** | prestige, coordination, image | prestige, organisational efficiency, coordination, image, quality of domain leadership | image, prestige |
| **Environmental values** | product composition, generation method (e.g., RES), environmental impact (RES), long-term environmental impact, capacity not to generate the carbon footprint | generation method (e.g., RES), environmental impact (RES), long-term environmental impact, capacity not to generate the carbon footprint | product composition, production method (e.g., RES), disposability, option to be processed and reused, environmental impact (RES), long-term environmental impact (e.g., proponents of ecological farming), capacity not to generate the carbon footprint |

An important advantage of innovative business models in renewable energy generation, vital for the growth of the contemporary economy, is that they constitute sustainable business models. This means that the value they generate is multidimensional and, in addition to technical and economic assets, offers vital social aspects. The sources of the created value are for the most part technical architecture elements, i.e., innovative energy generators, smart networks, metering and ICT devices. Of no less importance in the creation of value are digital economy systems and human capital competences enabling the execution of energy generation and distribution processes.

*5.5. Research Summary and Answers to Research Questions*

Comparative analysis of the examined business models, in terms of assessment of their applicability in specific economic and public sectors, has provided answers to the following questions:

**Q1: What are the most important antecedents for the development of innovative business models in the RES-based energy sector?**

A stimulus for the development of innovative (renewable) energy generation business models is the increase in the energy generated from renewable sources, which varies rather significantly with the particular RES installation. The highest growth can be seen in PV and wind generation, while the growth in hydroengineering is sluggish. However, what should be noted is a high growth rate and the need to intensify RES investment activities resulting from energy policy (both national and EU) targets. Poland's Energy Policy by 2040, adopted in 2021, responds to key challenges ahead and charts the directions of the

energy sector development, considering tasks that need to be implemented in terms of the technologies to build a low-emission energy system. The objectives of the adopted energy policy include energy security whilst ensuring competitiveness of the economy and improvement in its energy efficiency, along with reduction of the energy sector's impact on the environment, while making optimum use of the country's own energy resources. In the area of management, a symptom of the ongoing transition in energy generation is the change in existing business models and the development of novel models. The latter are largely based on innovations covering/relating to:

- new low- or zero-emission energy generation methods that use distributed, mostly renewable, energy sources,
- development of entirely new relationships, such as energy producer–energy distributor–prosumer,
- application of smart energy grids.

Solutions and projects that apply such business models are part of the energy system segment that is referred to as innovative energy generation. The energy segment as a concept covers, in addition to the energy sector, participants of the energy market, the R&D domain and the energy generation environment, which are important elements of the economy and the innovation ecosystem.

**Q2: What particular constituents and properties of business models can be identified in the examined cases?**

In the case of the technology for the conversion of sunlight into electricity (photovoltaic farms), the primary source of value is the application of RES. Value for the prosumer is the electricity produced that is used for their own needs and its excess is transferred to the energy undertaking for balancing, which helps ensure energy self-sufficiency (for electricity, where the total savings on electricity are EUR 27,467 (PLN 131,369) per year.

As is also the case with passive buildings, the primary source of value is the application of a RES solution and generation of energy savings. The effects achieved are primarily the value for tenants, comprising modern and safe interiors and a convenient location, where the capacity of the photovoltaic plant is sufficient to cover the annual demand for energy from the building's systems, such as heating, cooling and ventilation.

The agricultural biogas produced by the biogas plant, generated from local substrates, will also supply, as an alternative, a distributed co-generation module with 55 kWe/88 kWt in output. Value for residents of the commune is the generated electricity and heat used in public facilities. Reduction in the quantity of waste deposited in the environment will improve air quality and reduce $CO_2$ emissions. The improvement in energy security and energy efficiency of local public facilities and ensuring a backup source of supply for those facilities will enable the promotion of efficient energy use.

**Q3:How can the methodology be developed to examine business models in terms of value creation?**

Innovation-based value is the central dimension of the contemporary business model, and its sources are the various types of innovation, which is clearly relevant for the energy generation sector. Many discussions emphasise close links between the business model and value creation for customers and the business, which translates into profitability and competitiveness of the business.

Further research should focus on the improvement of the developed methodology. The methodology of business model research in terms of value creation should, at the first stage, cover analytical and background work to identify business models and analyse internal and external factors, so that, at the next stage, the structure and properties of such business models can be examined. Items of social and technical architecture and business process should be considered here.

## 6. Conclusions

The identification of antecedents for business models in renewable energy generation allows the argument that the business model can be treated as a system for the configuration

of resources and interdependent activities, emphasising the role of the configuration of tangible and intangible elements. The conclusion from the characterisation of the concepts and structures presented here is that the collection of such activities, resources, the method of their organisation and links between activities, resources and the value network, which enable such activities in cooperation with partners or customers, are clearly dependent on the adopted business model. The application of the methodology for research on the structure of business models in innovative energy generation has enabled the assessment of their capacity to create multidimensional value.

The circumstances, factors and reasons antecedent to the creation of business model concepts and structures are rooted in the development of the global economy, the associated innovation and competitiveness, and are driven by respect for the environment and human safety. External antecedents for business models come from the macroeconomic, social, legal and R&D environment. Internal antecedents, in turn, cover the situation and perspectives of the energy sector and undertakings and include the economic and ownership situation of the sector, its production structure (energy mix), personnel competence, financing and support for renewable energy generation, innovative energy sector technologies, and Economy 4.0 instruments. The antecedence of RES business models is not limited to their identification and characterisation, but also covers the exploration of concepts and structures of the emerging models. The presented RES business models demonstrate the relevance of such antecedents for their concepts and structural constituents. For example, RES energy generation is the result of climate policy, sustainable development, smart grid and smart metering, which are constituents of the model's technical architecture, whereas the application of Economy 4.0 instruments, passive building and biogas plant funding from support funds is an example of how instruments for the support of renewable energy generation are used. The presented innovative energy generation models show only a portion of the many solutions in the area of RES. They employ various social and technical architectures and business processes and create different values, generating different results. Comparative analysis of renewable energy generation business models indicates that what they share is a broad application of innovation, e.g., modern energy generators and ICT systems employing the instruments of Economy 4.0 (smart energy grids, artificial intelligence), and the use of renewable biological resources for the generation of energy. The examined models share the general antecedence for the creation of a product that is human-friendly and friendly to the ecosystem in which humans live. They constitute the so-called distributed energy sources that cooperate with regional electricity distribution system operators, which improves energy security even though only one of the examined models (the biogas plant) ensures continuous energy production. It should be noted that the presented business cases used external funding in the development of their business models.

Further growth of innovative (including renewable) energy generation will involve changes to and development of business models. The most important changes in renewable energy generation business models are the integration into energy generation of processes for its storage, and a new form of integrated co-generation, that is, the production of electricity, heat and cold. Therefore, a new business model architecture is emerging based on the source of electricity (photovoltaic sources, wind power and organic sources), heat (heat pumps, sun collectors), energy storage and new cybersystems that integrate the operation of energy equipment and enable collaboration with distribution systems. Key considerations in the development of such models are the ever more stringent climate policy and the requirement to provide load balancing for energy grids through the elimination of the weakness of renewable energy generation, that is, instability of generation. An important factor in the development of such business models is also a massive advance in the technologies for energy storage (galvanic and flow cells, water reservoirs, liquid and compressed air storages, heat tanks, hydrogen production). The development of energy storage technologies responds to the weakness of RES, i.e., instability in generation.

As the presented business models have been active for a relatively short time, changes in competence and human capital attitudes, as well as social acceptance of those models, are not examined here. The authors posit that there is a strong need to study the operation of wind farms. Clearly there is economic and technical acceptance of those models, but the authors are not aware of any in-depth studies on social acceptance in this respect. Statements by wind energy sector experts typically reveal the standpoint of investors and their lobbyists, leaving out the arguments of social partners. As the efficiency limit has not been determined, this could be another potential line of research, as the optimisation algorithm or system variables have no upper or lower limit value.

**Author Contributions:** Conceptualization, J.B., L.K. and M.O.-P.; Formal analysis, D.K.Z.; Investigation, J.B.; Methodology, J.B., L.K. and M.O.-P.; Resources, L.K.; Writing—original draft, M.O.-P.; Writing—review & editing, J.B., L.K., M.O.-P. and D.K.Z. All authors have read and agreed to the published version of the manuscript.

**Funding:** This research received no external funding.

**Institutional Review Board Statement:** Not applicable.

**Informed Consent Statement:** Not applicable.

**Data Availability Statement:** Not applicable.

**Conflicts of Interest:** The authors declare no conflict of interest.

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
