# Peer review of "Antecedents of Creating Business Models in the Field of Renewable Energy Based on the Concept of the New Age of Innovation"

_energies, doi:10.3390/en15155511_

Round 1
Reviewer 1 Report
This work talks about the innovative renewable energy business, which sounds interesting. However, the authors also need to consider the following questions.
1. The main contribution of the paper should be highlighted and emphasized. It would be great if the drawbacks and gaps of literature are clear and, particularly, how the proposed approach aims at filling these gaps.
2. The abstract should briefly display the results of the research.
3. It is better to make a more comprehensive literature review in the form of a table (matrix) so that the reader is more confident with the contribution of this research.
4. The performance required presenting in more quantitative manner
5. Improve the English writing/editing of the manuscript. There are many grammatical mistakes throughout the manuscript.
6. Add more results to validate the proposed work and compared those with the existing analysis/work. Moreover, the computational effort and accuracy of the proposed work should be compared with a benchmark method and other existing work to justify its effectiveness.
7. Explain in brief how the present paper differs from the published ones.
8. Present the proof of sensitivity and robustness formulations/analysis of the proposed work and validation.
9. It is necessary that the authors should illustrate/present the details of the proposed work, modeling/design and data of the studied power system, system constraints/data/parameters, etc. Moreover, state the system constraints, in other words, the upper and the lower boundaries of the optimization algorithm/system variables, etc.
10. What are the limitations and disadvantages of the proposed work?
Author Response
Dear Sir or Madam,
I’d kindly introduce you our corrections and supplements in the article “Antecedents of creating business models in the field of renewable energy based on the concept of the New Age of Innovation” after taking into account your comments. At the same time, we kindly thank you for your valuable suggestions.
In responding to the Comments and Suggestions for Authors addressed to us, on behalf of all the authors of this article, I present the improvements made (in the order of comments / suggestions of the reviewer):
- The main contribution of the paper should be highlighted and emphasized. It would be great if the drawbacks and gaps of literature are clear and, particularly, how the proposed approach aims at filling these gaps.
In line with the proposed suggestions - it was clearly highlighted in the article. The main contribution of the work was emphasized by presenting it in the abstract and in the introduction. The research gap in terms of identifying antecedents and their impact on the concepts and structural elements of the presented RES business models was clearly specified.
- The abstract should briefly display the results of the research.
The research results are presented briefly in the abstract.
- It is better to make a more comprehensive literature review in the form of a table (matrix) so that the reader is more confident with the contribution of this research.
The literature review in the form of a table was abandoned, but some items were omitted and amplified, ensuring the clarity and transparency of the argument.
- The performance required presenting in more quantitative manner
In line with the proposed suggestions - the research methodology was refined in the article to make it more explicit, specifying how the data was collected and how the proposed three steps were successfully implemented, presenting data from the implementation stages. Section 4. Research Methods and Materials presents an in-depth description of the author's own research. The desk research method used comes down to the analysis of the records from available data sources, including in particular their compilation, cross-verification and processing. The qualitative nature of the research was also emphasized.
- Improve the English writing/editing of the manuscript. There are many grammatical mistakes throughout the manuscript.
In line with the suggestion regarding the need to correct grammatical errors - the corrected manuscript was sent back for translation to another specialist.
- Add more results to validate the proposed work and compared those with the existing analysis/work. Moreover, the computational effort and accuracy of the proposed work should be compared with a benchmark method and other existing work to justify its effectiveness.
The proposed research has been reviewed with reference to the existing work, but it is probably not sufficiently exposed. This has been fixed by creating a separate point 3. Theoretical concept of research. The lack of a comparative method, the importance of the proposed method and the limitations of the research were also emphasized.
- Explain in brief how the present paper differs from the published ones.
In the conclusion, the originality of the conducted research was emphasized. A novelty is in the presentation of the concept of RES models and their structures - no in-depth research has been conducted in this area, or with the use of such methods.
- Present the proof of sensitivity and robustness formulations/analysis of the proposed work and validation.
The results of the research conducted were highlighted in the conclusion, they were also indicated in the abstract. Limitations are also indicated.
- It is necessary that the authors should illustrate/present the details of the proposed work, modeling/design and data of the studied power system, system constraints/data/parameters, etc. Moreover, state the system constraints, in other words, the upper and the lower boundaries of the optimization algorithm/system variables, etc.
Details of the description of the examined models have been made more precise - with regard to individual case studies and in the conclusions. The lack of upper and lower limits of the optimization algorithm or system variables for the achieved efficiency was indicated due to the relatively short period of operation of the presented models.
- What are the limitations and disadvantages of the proposed work?
Due to the relatively short period of operation of the presented models - the changes in competence and attitudes of human capital, as well as social acceptance for these models, have not been examined? The authors see in particular a need to conduct research on the functioning of wind farms, because while the economic and technical acceptance of the analyzed models is there, the authors are not aware of in-depth research in terms of social acceptance. Experts' statements on wind energy usually present the standpoint of investors and their lobbyists, leaving out the arguments of the public.
The achieved efficiency was also not investigated due to the lack of upper and lower limits of the optimization algorithm or system variables. This could be an interesting area for further research.
Yours sincerely,
Monika Odlanicka-Poczobutt

Reviewer 2 Report
1. Table 1, what does BOS, DOS, ICT, DER mean? Their meanings should be written for the first time of using the acronyms.
2. What CAPEX came from? line 253
3. The research methodology does not specify how the data will be collected and how the three steps will be effectively implemented. The authors should also present how the data from the three stages of implementation will be processed. The reader also is expected to find a quantitative analysis of the three business and software situations in which this is done. As it is the methodology is far too general, ambiguous.
4. The currency of expression of fixed assets should be in a more general currency. I suggest the authors to convert or write in parenthesis the euro and/or dollar equivalent of the value expressed in pln.
5. In my opinion the conclusion section is very large. I suggest to the authors that the answers to the suggested questions (Q1-Q3) be included in the discussion section.
6. It is not specified very clearly what gap from literature this research covers.
7. In the conclusion section it should be written more explicitly what are the limitations of the research and, of course, what are the future research.
Author Response
Dear Sir or Madam,
I’d kindly introduce you our corrections and supplements in the article “Antecedents of creating business models in the field of renewable energy based on the concept of the New Age of Innovation” after taking into account your comments. At the same time, we kindly thank you for your valuable suggestions.
In responding to the Comments and Suggestions for Authors addressed to us, on behalf of all the authors of this article, I present the improvements made (in the order of comments / suggestions of the reviewer):
- Table 1, what does BOS, DOS, ICT, DER mean? Their meanings should be written for the first time of using the acronyms.
The meanings of the acronyms in Table 1 and throughout the text have been added (such as BOS, DOS, ICT) and they are given in italics when the acronyms are used for the first time. They were probably removed by the translator (the manuscript after the corrections was sent once more for translation).
- What CAPEX came from? line 253
CAPEX means capital expenditure on product development or system implementation to maintain the company's current capacity to generate income, but it was not the intention of the authors to refer to this concept (translator's case), therefore it was removed. The explanation in the text of this issue in relation to the ESCO models seems to be sufficient.
- The research methodology does not specify how the data will be collected and how the three steps will be effectively implemented. The authors should also present how the data from the three stages of implementation will be processed. The reader also is expected to find a quantitative analysis of the three business and software situations in which this is done. As it is the methodology is far too general, ambiguous.
In line with the proposed suggestions - the research methodology was refined in the article to make it more explicit and specify how the data was collected and how the proposed three steps were successfully implemented, with the presentation of data from the implementation stages.
- The currency of expression of fixed assets should be in a more general currency. I suggest the authors to convert or write in parenthesis the euro and/or dollar equivalent of the value expressed in pln.
The values of fixed assets were converted and expressed in euros, it was indeed an oversight..
- In my opinion the conclusion section is very large. I suggest to the authors that the answers to the suggested questions (Q1-Q3) be included in the discussion section.
As suggested, the answers to the questions (Q1-Q3) have been included in the discussion section.
- It is not specified very clearly what gap from literature this research covers.
The research gap in terms of identifying antecedents and their impact on the concepts and structural elements of the presented RES business models was clearly specified. The main contribution of the work was presented in the abstract and in the introduction.
- In the conclusion section it should be written more explicitly what are the limitations of the research and, of course, what are the future research.
The conclusion section has been amplified with research limitations and suggestions for future research. As the presented business models have been active for a relatively short time, changes in the competence and human capital attitudes, as well as social acceptance for those models, are not examined here. The authors posit that there is a strong need to study the operation of wind farms. Clearly there is economic and technical acceptance of those models, but the authors are not aware of any in-depth studies on social acceptance in this respect. Statements by wind energy sector experts typically reveal the standpoint of investors and their lobbyists, leaving out arguments of social partners.
As the efficiency limit has not been determined, it can be another potential line of research, as the optimization algorithm or system variables have no upper or lower limit value.
Yours sincerely,
Monika Odlanicka-Poczobutt

Round 2
Reviewer 1 Report
The authors have made sufficient modifications according to the modification comments.
Reviewer 2 Report
From my point of view the paper should be accepted in the present form.